# Sex prediction through machine learning utilizing mandibular condyles, coronoid processes, and sigmoid notches features

**Isabela Bittencourt Basso**[1], **Pedro Felipe de Jesus Freitas**[2], **Aline Xavier Ferraz**[3,4],
**Ana Julia Borkovski**[5], **Ana Laura Borkovski**[5], **Rosane Sampaio Santos**[3,4],
**Rodrigo Nunes Rached**[1], **Erika Calvano Küchler**[6], **Angela Graciela Deliga Schroder**[2,3,4],
**Cristiano Miranda de Araujo**[2,3,4], **Odilon Guariza-Filho**[1,4] *

1 Postgraduate Program in Dentistry, Pontifícia Universidade Católica do Paraná, Curitiba, Brazil, 2 School of Dentistry, Tuiuti University of Paraná, Curitiba, Paraná, Brazil, 3 Postgraduate Program in Human Communication Health, Tuiuti University of Paraná, Curitiba, Paraná, Brazil, 4 Center for Artificial Intelligence in Health–NIAS, Curitiba, Paraná, Brazil, 5 Graduate Program in Dentistry, Pontifícia Universidade Católica do Paraná, Curitiba, Brazil, 6 Medical Faculty, Department of Orthodontics, University Hospital Bonn, Bonn, Germany

* odilongfilho@gmail.com

**Data Availability Statement:** The data underlying the results of this study are available in the

## Abstract

Characteristics of the mandible structures have been relevant in anthropological and forensic studies for sex prediction. This study aims to evaluate the coronoid process, condyle, and sigmoid notch patterns in sex prediction through supervised machine learning algorithms. Cephalometric radiographs from 410 dental records of patients were screened to investigate the morphology of the coronoid process, condyle, and sigmoid notch and the Co-Gn distance. The following machine learning algorithms were used to build the predictive models: Decision Tree, Gradient Boosting Classifier, K-Nearest Neighbors (KNN), Logistic Regression, Multilayer Perceptron Classifier, Random Forest Classifier, and Support Vector Machine (SVM). A 5-fold cross-validation approach was adopted to validate each model. Metrics such as area under the curve (AUC), accuracy, recall, precision, and F1 Score were calculated for each model, and ROC curves were constructed. All tested variables demonstrated statistical significance (p < 0.10) and were included in the construction of the predictive model. The Co-Gn variable stood out as the most important among the evaluated independent variables, showing greater relevance in three of the four algorithms used in assessing feature importance. In the analysis of the models' performance, the AUC ranged from 0.82 [95% CI = 0.72–0.93] to 0.66 [95% CI = 0.53–0.76] for the test data, and from 0.83 [95% CI = 0.80–0.87] to 0.71 [95% CI = 0.61–0.75] for cross-validation. The precision of the models ranged from 0.83 [95% CI = 0.75–0.91] to 0.68 [95% CI = 0.58–0.78] in the test phase, and from 0.78 [95% CI = 0.74–0.82] to 0.69 [95% CI = 0.65–0.75] in cross-validation. The SVM, KNN, and Gradient Boosting Classifier algorithms stood out with the highest AUC and precision values in both cross-validation and testing. The use of condyle, coronoid process, and sigmoid notch characteristics, in combination with supervised machine learning predictive models, shows potential for contributing to sex prediction based on morphometric

repository at the following doi: 10.6084/m9.
figshare.26835760.

**Funding:** The author(s) received no specific
funding for this work.

**Competing interests:** The authors have declared
that no competing interests exist.

bone characteristics, particularly regarding the distance between the condyle and gnathion.
However, given the study's limitations, these findings should be interpreted with caution.

## Introduction

Sex identification is one of the central objectives in forensic science for determining the biological profile, along with age, ancestry, and stature. In the absence of soft tissues, forensic professionals can rely on the analysis of skeletal remains for identification [1]. This process involves the examination of specific anatomical areas of the bones, due to their preservation after death. In forensic odontology, skull analysis is frequently conducted, given its significant importance in determining sexual identity [2,3]. The mandible, is the most robust part of the skull, exhibits considerable variation between sexes, commonly presenting a larger size in the male mandible compared to the female [4].

Additionally, other structures of the mandible, such as the condyle, coronoid process, and sigmoid notch, may also display dimorphic characteristics [5]. Morphological variations of the coronoid process, condyle, and sigmoid notch have been used as important points in anthropological and forensic studies. Muscle activity in the condylar region can modify the shape of the condyle and coronoid process over time. However, genetic factors and hormonal fluctuations also play a role in shaping the sigmoid notch, thus highlighting sexual dimorphism [5–7].

Technological advancements, particularly artificial intelligence models, have been able to solve problems through logical reasoning, thus becoming an important auxiliary tool in decision-making. In forensic odontology, this technology has shown promise, providing reliable data for professionals to make informed decisions, enabling the identification of patterns and trends that may not be immediately perceptible [8]. Several studies have evaluated mandibular measurements for sex determination, including the dimensions of the mandibular body and ramus, gonia-menton distance, bigonial, bicondylar, and bicoronoid widths, coronoid-menton distance, mandibular lingula, gonial angle, as well as the height and length of the condyles and coronoids. These studies have demonstrated strong predictive capacity when supervised machine learning techniques are applied [9–12]. In particular, linear measurements of the height of the coronoid and condyle have shown high predictive power [10].

However, no studies have been identified that exclusively explore the use of supervised machine learning techniques applied solely to the linear morphometric measurements of the condyle, coronoid process, and sigmoid notch for sex prediction. Most available studies integrate these features with other mandibular measurements, addressing them in a combined manner. This study aims to fill this gap by evaluating whether the characteristics of the condyle, coronoid process, and sigmoid notch can independently provide accurate sex predictions using supervised machine learning algorithms. This focused approach may simplify the data collection process and enhance the applicability of the results in specific forensic contexts, particularly when other mandibular features are missing or degraded.

Therefore, the aim of this study is to evaluate the use of condyle, coronoid process, and sigmoid notch characteristics in sex prediction, using supervised machine learning algorithms.

## Materials and methods

### Study design and setting

This cross-sectional study analyzed 410 dental records of patients at a dental radiology center located in the southern region of Brazil. The research was conducted following the principles

of the Helsinki Declaration and received approval from the Research Ethics Committee of Tuiuti University of Paraná, Brazil (Approval Number: 6.305.456), including administrative permission to access the data used in this study, and waiver of informed consent as long as it did not imply patient identification. Thus, all data were anonymized before being accessed. Access to all data occurred in March 2024.

## Participants

The following eligibility criteria were established for sample composition: patients over 10 years old, with no history of surgery related to any area of the mandible, of both sexes. Patients with facial malformations, syndromes, or a history of severe facial trauma were excluded from the sample.

## Study variables and data collection

Data collection was conducted by two assessors and involved the use of cephalometric tracings made on lateral cephalograms and panoramic radiographs. All tracings were performed by an orthodontist specialized and master's degree holder in Dental Radiology. All exams were executed using the Orthophos S 2D/3D equipment from Dentsply Sirona (Dentsply Sirona Inc, New York, USA). The following variables were collected:

a. Mandibular condyle morphology - Defined based on the shape of the condylar process, categorized as round, bird beak, crooked finger, diamond, or flat (Fig 1A). This classification was obtained through panoramic radiographic examination, following specific criteria outlined in previous studies [5,6,13]. The assessment focused on the most prominent points of the superior, anterior, and posterior contours of the condyle.

b. Coronoid process morphology - Defined by the shape of the coronoid process, categorized as round, triangular, crooked finger, flat, or beak (Fig 1B). This definition was based on panoramic radiographs and followed morphological descriptions established in the literature [5,6,13]. The shape was determined by analyzing the most prominent points along the border of the coronoid process.

c. Sigmoid notch morphology - Described by the shape of the sigmoid notch, categorized as round, sloping, V-shaped, or flat (Fig 1C). The definition was based on the curvature between the condyle and the coronoid process, using panoramic radiographic images and criteria from previous studies [5,6,13]. The points used to evaluate the sigmoid notch considered the ends and the maximum depth of the curvature.

d. Co-Gn distance - The distance in millimeters between the gnathion (Gn) point and the condyle (Co), measured from the most superior and posterior point of the mandibular condyle to the most anterior and inferior point of the mandible. The measurement was performed using cephalometric tracings on lateral cephalograms.

The classification of condyle, coronoid process, and sigmoid notch morphology followed criteria previously established by studies described in the literature [5,6,13]. To ensure the reproducibility of both intra- and inter-examiner measurements, ten measurements were randomly selected, which were reassessed after one week, resulting in Kappa coefficient and Intraclass Correlation Coefficient values exceeding 0.8 for both intra- and inter-examiner assessments.

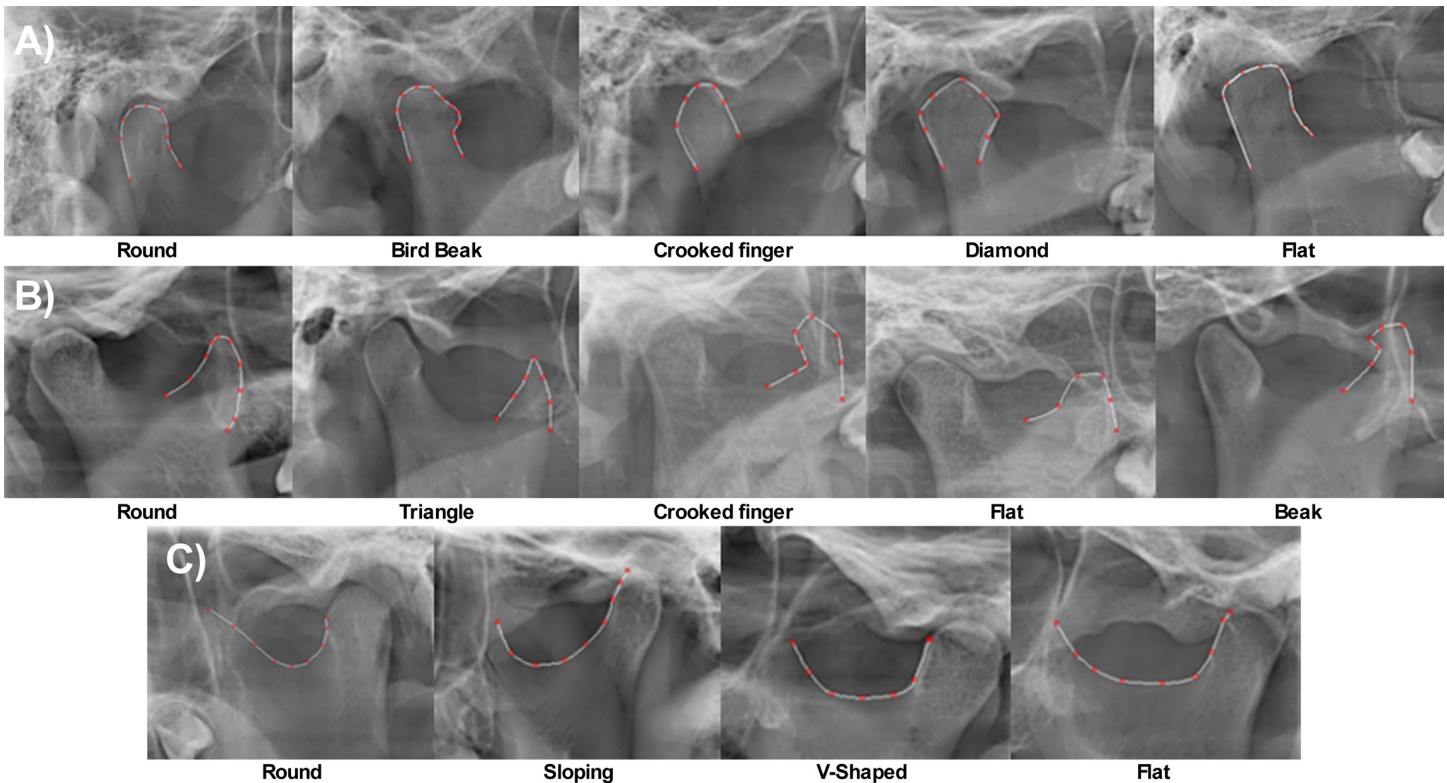

**Fig 1.** Condyle (A), coronoid process (B), and sigmoid notch (C) morphology.

### Data analysis and model construction

To ensure the selection of variables with the highest predictive power, a univariate analysis was conducted using the chi-square test for categorical variables and the Student's t-test for the Co-Gn variable. A significance level of 10% ($\alpha = 0.10$) was adopted to allow the inclusion of any relevant variable for the model. A significance level of 10% ($\alpha = 0.10$) was adopted to allow the inclusion of any potentially relevant variable for the model. This significance level was chosen to set a conservative cutoff point, thereby preventing the premature exclusion of any variable with potential predictive importance. However, to avoid the inclusion of a suboptimal set of variables, the recursive feature elimination with cross-validation (RFECV) technique was also employed, using the 'RFECV' class from the 'scikit-learn' library in Python. These approaches aimed to select the most relevant features to achieve the optimal subset of characteristics, optimizing the performance of the machine learning algorithms [14,15]. The RFECV works by iteratively evaluating the importance of features, removing the least important ones, and recalculating the model's performance in each iteration. This is done using estimators that have an associated feature importance measure, such as tree-based or linear estimators. Thus, RFECV helps identify which features contribute most to the predictive capability of the model.

To address class imbalance in the training set, the Synthetic Minority Over-sampling Technique (SMOTE) was employed [16]. This oversampling approach synthesizes examples from the minority class, increasing its representation in the training data. SMOTE was exclusively applied to the training set using the 'imbalanced-learn' library in Python to avoid leakage of information from the test set. This procedure contributed to improving the robustness of the model, especially regarding less represented classes.

The following machine learning algorithms were used to build the predictive models: Decision Tree, Gradient Boosting Classifier, K-Nearest Neighbors (KNN), Logistic Regression, Multilayer Perceptron Classifier (MLP), Random Forest Classifier, and Support Vector Machine (SVM). Model performance optimization was performed using the Grid Search method, thus finding the best combination of hyperparameters for each algorithm. All analyses were conducted using the Python programming language, associated with the 'scikit-learn' library (DOI: 10.5281/zenodo.11077427).

## Training, cross-validation, and test

Initially, the data was split into training and cross-validation sets, with 80% of the data designated for training and validation, while the remaining 20% was exclusively reserved for testing the predictive capacity of the model. For this purpose, the 'train_test_split' function from the 'sklearn.model_selection' library was used.

The evaluation of the model's generalization ability to unseen data was performed using cross-validation technique. In this approach, the data was divided into k subsets, and the model was trained k times. In each iteration, k-1 subsets were used for training, while the remaining subset was reserved for validation. This allowed calculating an average estimate of validation performance, using a 5-fold cross-validation.

## Metrics and model evaluation

The discriminative ability of the models regarding different prediction classes was evaluated using the Receiver Operating Characteristic (ROC) curve. For each model, the area under the curve (AUC) was calculated using the 'roc_auc_score' function from the 'scikit-learn' package, providing a quantitative measure of discriminative power. This analysis involved determining the false positive and true positive rates at various classification thresholds using the 'roc_-curve' function from the 'scikit-learn' package, with the actual labels from the test set and the predicted probabilities for the positive class of each model. ROC curve plots were generated using the 'matplotlib.pyplot' library. Additionally, accuracy, recall, precision, and F1 Score metrics were calculated for each model. These metrics were chosen based on their use in similar studies [3,9–12], allowing for a more meaningful comparison of the model's performance in sex prediction.

- Accuracy: indicates the model's overall performance (accuracy rate), determined by the ratio of correct predictions to the total number of instances.

$$Accuracy = \frac{(True\ positives + True\ negatives\ )}{\text{Total elements}}$$

- Precision: measures the correctness of the model's positive predictions by calculating the proportion of true positives among all the positive predictions it makes.

$$precision\ = \frac{True\ positives}{\text{True positives} + \text{False positives}}$$

- Recall: indicates the model's effectiveness in accurately detecting all instances of the actual positive class.

$$Recall = \frac{True\ positives}{\text{True positives} + \text{False negatives}}$$

- F1 Score: Provides a comprehensive assessment of the model by computing the harmonic mean of precision and recall.

$$F1\ Score = 2*\frac{precision * recall}{\text{precision} + \text{recall}}$$

The 95% confidence intervals (CI95%) were computed for all metrics obtained from both the test data and cross-validation. To calculate the CI95% for the test data, the bootstrap technique was employed, involving 1,000 iterations with random sampling with replacement. The CI95% was defined by the 2.5th and 97.5th percentiles of the bootstrap distribution of the metrics. During 5-fold cross-validation, the model was evaluated multiple times on different training and test subsets, with the same metrics being calculated for each subset. The CI95% was derived directly from the metrics obtained in each fold, providing an estimate of the variability in the model's performance across different data partitions.

To better understand which features influence the predictive capability of each model the most, the feature importance evaluation function from the 'scikit-learn' library was used. This visual analysis allowed identifying the most relevant variables in each model formulation. However, this evaluation could not be conducted for the KNN, SVM, and MLP models due to the peculiarities of these algorithms, which do not support this function.

## Results

Out of the 410 patient records evaluated, 74 were excluded due to the age (under 10 years), resulting in 336 patients included in the analysis. The average age of the sample was 23.7 ± 13.5 years for males and 25.3 ± 15.7 for females, with the sample divided into 41.4% males and 58.6% females. To ensure the representativeness of the groups, oversampling technique was employed to balance the sample. The steps of the predictive model construction process are illustrated in Fig 2.

All tested variables showed statistical significance ($p < 0.10$) and were initially included in the construction of the predictive model (Table 1). After the application of the RFECV technique, only in the logistic regression model, two variables related to the morphometry of one side of the sigmoid and condyle were eliminated, aiming to achieve a more precise predictive fit. The Co-Gn variable stood out as the most important among the evaluated independent variables, showing greater relevance in three out of the four algorithms used in assessing feature importance (Fig 3). When analyzing the performance of the models, it was found that the AUC ranged from 0.82 [95% CI = 0.72–0.93] to 0.66 [95% CI = 0.53–0.76] for the test data, and from 0.83 [95% CI = 0.80–0.87] to 0.71 [95% CI = 0.61–0.75] for cross-validation (Fig 4). The precision of the models ranged from 0.83 [95% CI = 0.75–0.91] to 0.68 [95% CI = 0.58–0.78] in the test phase, and from 0.78 [95% CI = 0.74–0.82] to 0.69 [95% CI = 0.65–0.75] in cross-validation. The SVM, KNN, and Gradient Boosting Classifier algorithms stood out with the highest AUC and precision values in both cross-validation and testing. The detailed performance of all predictive models can be found in Table 2.

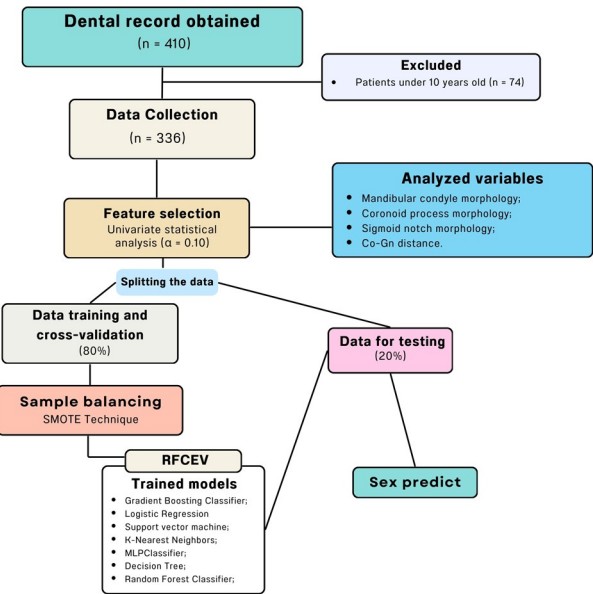

**Fig 2. Flowchart diagram illustrating the data analysis process using machine learning models.**

## Discussion

One of the main goals of forensic anthropology is to recreate the biological profile of deceased individuals, which involves estimating sex from skeletal remains [17]. Bone morphology emerges as a viable method and, in some cases, the only option, as it allows for swift analysis, with reduced cost and based solely on anatomical characteristics, without damaging or consuming the sample [18]. In this context, the present study aimed to evaluate the use of condyle, coronoid process, and sigmoid notch characteristics in sex prediction using supervised machine learning algorithms. However, to date, no studies have been found that explore machine learning approaches using exclusively condyle, coronoid process, and sigmoid notch characteristics, whether morphometric or linear, for sex prediction.

Craniofacial morphology is influenced by genetic factors [6]. The human mandible, as the most robust bone of the face, dynamically interacts with the masseter muscle, inserting into the structures of the mandibular ramus and the zygomatic bone to perform its movements. Mandibular morphological variation plays a crucial role in assessing facial characteristics and determining the individual's facial growth pattern, besides it is an important indicator of age, ethnicity, and sex, often used in anthropological and forensic investigations [6,19]. Other studies have employed different mandibular structures for sex prediction, combined with supervised machine learning models, achieving strong model performance metrics [9,11,12]. Peterk et al. (2024) used mandibular morphometric measurements obtained from panoramic radiographs to determine sex through machine learning, achieving an accuracy of 0.82 when 12 variables were included, and 0.86 when only the height of the coronoid, height of the ramus, minimum width of the ramus, and bigonial width were used [10]. Similarly, Baban et al. (2023) achieved an accuracy of 0.90 using volumetric and linear measurements from cone-beam computed tomography (CBCT) images combined with machine learning techniques, when 13 variables were included [9]. In the present study, a smaller number of variables were included, focusing exclusively on the characteristics of the condyle, coronoid process, and sigmoid notch, resulting in an accuracy of 0.83 in the test phase. Although a reduced number of variables may decrease the model's accuracy, it enhances its applicability in real-world

**Table 1. Measurements by sex.**

| Measurement | Classification | n (%) / Mean ± SD | p-value* |
|---|---|---|---|
| Morphology of the mandibular condyle | | | |
| *Male* | Round | 147 (52.9%) | 0.029 |
| | Bird Beak | 70 (25.2%) | |
| | Crooked Finger | 10 (3.6%) | |
| | Diamond | 31 (11.2%) | |
| | Flat | 20 (7.2%) | |
| *Female* | Round | 230 (58.4%) | |
| | Bird Beak | 67 (17.0%) | |
| | Crooked Finger | 12 (3.0%) | |
| | Diamond | 65 (16.5%) | |
| | Flat | 20 (5.1%) | |
| Morphology of the coronoid process | | | |
| *Male* | Round | 153 (55.0%) | 0.018 |
| | Triangle | 109 (39.2%) | |
| | Crooked Finger | 13 (4.7%) | |
| | Flat | 0 (0.0%) | |
| | Beak | 3 (1.1%) | |
| *Female* | Round | 170 (43.1%) | |
| | Triangle | 206 (52.3%) | |
| | Crooked Finger | 16 (4.1%) | |
| | Flat | 1 (0.3%) | |
| | Beak | 1 (0.3%) | |
| Morphology of the sigmoid notch | | | |
| *Male* | Round | 46 (16.5%) | 0.086 |
| | Sloping | 73 (26.3%) | |
| | V-Shaped | 88 (31.7%) | |
| | Flat | 71 (25.5%) | |
| *Female* | Round | 81 (20.6%) | |
| | Sloping | 105 (26.6%) | |
| | V-Shaped | 139 (35.3%) | |
| | Flat | 69 (17.5%) | |
| Co-Gn | | | |
| *Male* | | 114 ± 8.16 | < 0.001 |
| *Female* | | 107 ± 5.48 | |

*p-value of the chi-square test for categorical variables and Student's t-test for independent samples for continuous variables.

scenarios, where few structures may be available for sex identification in forensic contexts. No other studies were found that specifically evaluated the predictive power of these structures alone using machine learning.

Morphological variations associated with condyle shape and sigmoid notch width play an important role in anthropological and forensic investigations [6]. When exploring sexual dimorphism, it is observed that older men tend to have angular condyles, while women and younger individuals often exhibit condyles with a more rounded shape. This distinction can also be influenced by ethnicity, as the round or oval shape is common in both sexes in certain populations [5]. In the present study, individuals over the age of ten were included because many criminal investigations involve victims from various age groups, and it is crucial that

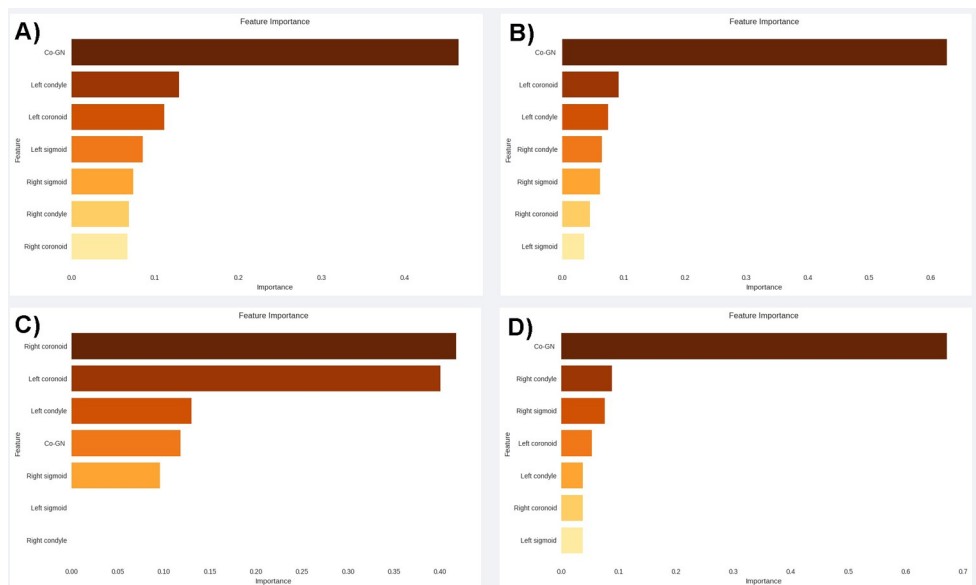

**Fig 3. Results of feature importance analysis from four machine learning models.** A–Decision Tree, B–Gradient Boosting Classifier, C–Logistic Regression, D–Random Forest Classifier.

identification and analysis methods be applicable to a wide age range. Moreover, age estimation based on the available structures may not provide accurate results in some scenarios where the structures are limited or of poor quality. Although this decision may introduce some variability in the results, especially regarding the accuracy of the algorithms used, the inclusion of a broad age range, spanning from childhood to adulthood, strengthens the model's ability to provide accurate estimates in cases of identifying remains or analyzing bone evidence. On the other hand, regarding ethnicity, additional studies with samples from other

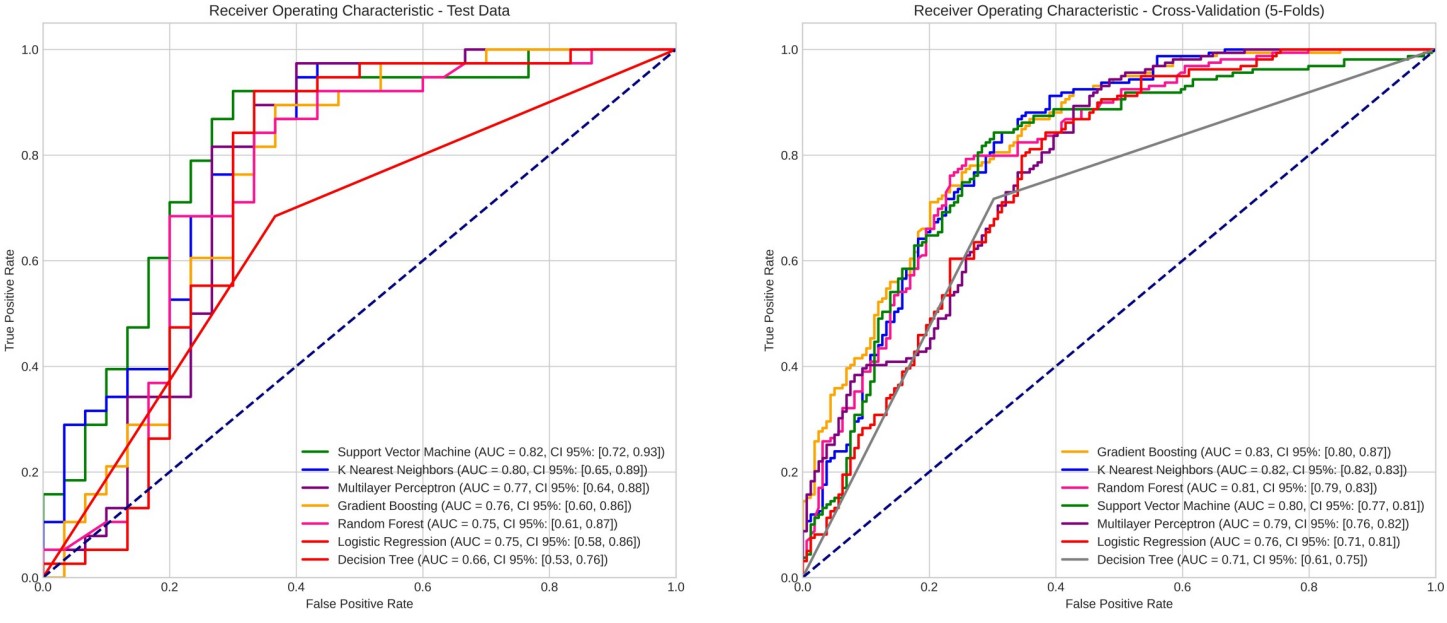

**Fig 4. Evaluation of classification models using ROC curves.**

**Table 2. Summary of metrics obtained for the cross-validation and test stages of the models, along with their respective optimal hyperparameters.**

| Model | Optimal Hyperparameters (random_state = 24) | Cross-validation results [CI95%] | Test Data Results [CI95%] |
|---|---|---|---|
| Logistic Regression | C: 1 | Accuracy = 0.69 [0.64–0.75] | Accuracy = 0.76 [0.66–0.85] |
| | max_iter: 100 | Precision = 0.69 [0.65–0.75] | Precision = 0.76 [0.66–0.86] |
| | penalty: l1 | Recall = 0.69 [0.64–0.75] | Recall = 0.76 [0.66–0.85] |
| | l1_ratio: 0.2 | F1-Score = 0.69 [0.64–0.75] | F1-Score = 0.76 [0.66–0.85] |
| | solver: liblinear | | |
| Gradient Boosting Classifier | n_estimators: 2000 | Accuracy = 0.76 [0.73–0.80] | Accuracy = 0.76 [0.66–0.87] |
| | learning_rate: 0.1 | Precision = 0.76 [0.74–0.81] | Precision = 0.77 [067–0.87] |
| | criterion: friedman_mse | Recall = 0.76 [0.73–0.80] | Recall = 0.76 [0.66–0.87] |
| | max_depth: 5 | F1-Score = 0.76 [0.73–0.80] | F1-Score = 0.76 [0.65–0.86] |
| | loss: exponential | | |
| K-Nearest Neighbors | n_neighbors: 100 | Accuracy = 0.76 [0.72–0.81] | Accuracy = 0.76 [0.66–0.85] |
| | weights: distance | Precision = 0.78 [0.74–0.82] | Precision = 0.77 [0.67–0.86] |
| | leaf_size: 1 | Recall = 0.76 [0.72–0.81] | Recall = 0.76 [0.66–0.85] |
| | p: 1 | F1-Score = 0.76 [0.71–0.81] | F1-Score = 0.76 [0.65–0.85] |
| Support Vector Machine | kernel: rbf | Accuracy = 0.76 [0.75–0.78] | Accuracy = 0.82 [0.74–0.91] |
| | C: 1.5 | Precision = 0.77 [0.75–0.79] | Precision = 0.83 [0.75–0.91] |
| | gamma: auto | Recall = 0.76 [0.75–0.78] | Recall = 0.82 [0.74–0.91] |
| | | F1-Score = 0.76 [0.75–0.78] | F1-Score = 0.82 [0.72–0.91] |
| MLP Classifier | activation: logistic | Accuracy = 0.71 [0.66–0.76] | Accuracy = 0.78 [0.68–0.87] |
| | alpha: 0.1 | Precision = 0.72 [0.68–0.76] | Precision = 0.78 [0.69–0.88] |
| | hidden_layer_sizes: 1000 | Recall = 0.71 [0.66–0.76] | Recall = 0.78 [0.68–0.87] |
| | learning_rate_init: 0.01 | F1-Score = 0.71 [0.66–0.76] | F1-Score = 0.78 [0.66–0.87] |
| | max_iter: 100 | | |
| | solver: lbfgs | | |
| Decision Tree | criterion: gini | Accuracy = 0.71 [0.62–0.76] | Accuracy = 0.68 [0.57–0.78] |
| | max_depth: 15 | Precision = 0.71 [0.62–0.76] | Precision = 0.68 [0.58–0.78] |
| | splitter: random | Recall = 0.71 [0.62–0.76] | Recall = 0.68 [0.57–0.78] |
| | | F1-Score = 0.71 [0.61–0.76] | F1-Score = 0.68 [0.57–0.78] |
| Random Forest Classifier | max_depth: None | Accuracy = 0.76 [0.74–0.78] | Accuracy = 0.75 [0.65–0.85] |
| | n_estimators: 5 | Precision = 0.76 [0.74–0.79] | Precision = 0.76 [0.65–0.86] |
| | min_samples_split: 15 | Recall = 0.76 [0.74–0.78] | Recall = 0.75 [0.65–0.85] |
| | min_samples_leaf: 6 | F1-Score = 0.76 [0.74–0.78] | F1-Score = 0.74 [0.64–0.85] |
| | criterion: entropy | | |
| | max_features: auto | | |

nationalities are needed to allow for a more comprehensive assessment in different ethnic groups.

The presence of sexual dimorphism in the temporomandibular joint muscle attachment morphometry results in mechanical differences in the masticatory system between men and women, which influence mandibular size [20]. In this context, discrepancies in the insertion and function of the masticatory muscles, as well as variations in occlusal load, hormone levels, and genetic factors, may play roles in modulating the morphology of the coronoid process, condyle, and sigmoid notch, varying both between sexes and between the right and left sides [5]. In the present study, the morphology of these structures was individually integrated into the model, allowing for a detailed and comprehensive analysis. Despite the inclusion of all variables in the model, it was observed that the distance between the condyle and gnathion was the most important variable. This may be attributed to the numerical nature of this variable,

which provides a substantial amount of information to the model, making it more informative for prediction.

Imbalanced data are common in supervised machine learning classification tasks, occurring when there is an unequal distribution between classes, which can distort the overall accuracy of the model, especially when the imbalance is pronounced [21]. A widely used technique to address this issue is SMOTE, which generates new data for the minority class by creating points along the lines connecting existing examples to their nearest neighbors. This technique is popular due to its simplicity and effectiveness. However, the accuracy of SMOTE is influenced by the sample size, with better precision in generating synthetic data for larger samples [22]. Although SMOTE may produce data that do not perfectly match the true distribution, in this study, the technique was applied to a large sample with no significant class imbalance. Furthermore, the metrics obtained from 5-fold cross-validation were very similar to those from the test phase, suggesting that the model did not suffer from overfitting. This consistency indicates a reduced risk of distortion and a strong capacity for generalization.

Some limitations of this study should be considered. The analysis focused specifically on morphological characteristics of the mandible, overlooking other craniofacial bone structures that may also play a role in sex prediction. Despite the high reproducibility achieved in the present study, morphometric analysis can be considered somewhat subjective. Although the characteristics of the condyle, coronoid process, and sigmoid notch have demonstrated significant predictive power, the exclusion of other structures may limit the breadth and accuracy of the developed predictive models. Additionally, it is important to note that, although the sample represented a mixed population, it was limited to a single ethnicity, consisting of Brazilian individuals. This may introduce biases and limitations in the applicability of the results to different ethnic groups. However, the use of these mandibular characteristics can complement the analysis of other structures, offering promising results for the forensic field. This was also the first study to specifically evaluate these structures in association with the use of supervised machine learning in a Brazilian population. The accuracy and utility of this approach make it a valuable addition to identification techniques, especially because it relies on a few mandibular structures, simplifying the analysis process in forensic contexts. To ensure the external validity of this model, it is recommended that it be evaluated in real-world scenarios and across diverse populations before being applied in forensic practice.

## Conclusions

The use of condyle, coronoid process, and sigmoid notch characteristics, together with supervised machine learning predictive models, shows potential for contributing to sex prediction based on morphometric bone characteristics, particularly concerning the distance between the condyle and gnathion. However, given the study's limitations, such as the focus on a single ethnic group and the exclusion of other craniofacial structures, these findings should be interpreted cautiously. Further validation in diverse populations and real-world forensic contexts is necessary to confirm the utility and generalizability of this approach.

## Author Contributions

**Conceptualization:** Aline Xavier Ferraz, Rosane Sampaio Santos, Cristiano Miranda de Araujo, Odilon Guariza-Filho.

**Data curation:** Ana Julia Borkovski, Ana Laura Borkovski, Cristiano Miranda de Araujo.

**Formal analysis:** Angela Graciela Deliga Schroder, Cristiano Miranda de Araujo.

**Funding acquisition:** Erika Calvano Küchler.

**Investigation:** Pedro Felipe de Jesus Freitas, Aline Xavier Ferraz, Ana Julia Borkovski, Ana Laura Borkovski, Cristiano Miranda de Araujo.

**Methodology:** Isabela Bittencourt Basso, Pedro Felipe de Jesus Freitas, Ana Julia Borkovski, Ana Laura Borkovski, Rosane Sampaio Santos, Rodrigo Nunes Rached, Erika Calvano Küchler.

**Software:** Angela Graciela Deliga Schroder, Cristiano Miranda de Araujo.

**Supervision:** Angela Graciela Deliga Schroder, Odilon Guariza-Filho.

**Visualization:** Rodrigo Nunes Rached.

**Writing – original draft:** Isabela Bittencourt Basso, Pedro Felipe de Jesus Freitas, Aline Xavier Ferraz.

**Writing – review & editing:** Rosane Sampaio Santos, Rodrigo Nunes Rached, Erika Calvano Küchler, Cristiano Miranda de Araujo, Odilon Guariza-Filho.

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
