## [Decision Letter · Decision Letter 0]

22 Aug 2024

PONE-D-24-17176SEX PREDICTION THROUGH MACHINE LEARNING UTILIZING MANDIBULAR CONDYLES, CORONOID PROCESSES, AND SIGMOID NOTCHES FEATURESPLOS ONE

Dear Dr. Guariza-Filho,

Thank you for submitting your manuscript to PLOS ONE. After careful consideration, we feel that it has merit but does not fully meet PLOS ONE’s publication criteria as it currently stands. Therefore, we invite you to submit a revised version of the manuscript that addresses the points raised during the review process.

In particular, reviewers have pointed out some lacks in the Introduction section that should be improved. Reviewers also recommended to increase the accuracy of the Methods section for what concern the metrics description. Discussion should better highlight limitations of the study, including also a more diffuse comparison with existing works.

The authors are required to carefully take into account all the reviewers' suggestions.

We look forward to receiving your revised manuscript.

Kind regards,

Giulia Pascoletti, Ph.D.

Academic Editor

PLOS ONE

Reviewers' comments:

Reviewer's Responses to Questions

**Comments to the Author**

1. Is the manuscript technically sound, and do the data support the conclusions?

Reviewer #1: Yes

Reviewer #2: Partly

Reviewer #3: Yes

2. Has the statistical analysis been performed appropriately and rigorously? 

Reviewer #1: I Don't Know

Reviewer #2: No

Reviewer #3: Yes

3. Have the authors made all data underlying the findings in their manuscript fully available?

Reviewer #1: No

Reviewer #2: No

Reviewer #3: Yes

4. Is the manuscript presented in an intelligible fashion and written in standard English?

Reviewer #1: Yes

Reviewer #2: Yes

Reviewer #3: Yes

5. Review Comments to the Author

Reviewer #1: The authors present a study focused on sex identification based on morphometric bone characteristics of the mandible.

Below my comments:

- the Introduction section is too concise in my opinion. An overview of the methodologies adopted by other studies with the same objective should be added, as well as on the adopted ML techniques.

- In the Methods section I strongly recommend to be more accurate in explaining how the morphometric variables later employed were defined, also with the help of images

- Did you check collinearity among the considered variables?

- As the Introduction, the Discussion could contain a broader comparison with other similar studies.

Reviewer #2: The introduction of the manuscript is quite inadequate. a more detailed literature review would have been expected from the authors.

The authors state "However, so far, no studies have been found in the literature that explore machine learning approaches using characteristics of the condyle, coronoid process, and sigmoid notch, whether morphometric or linear, for sex prediction." However, many features of mandible sex prediction have been successfully used and many ML models have been successfully introduced to the literature. This sentence requires serious clarification.

Although the authors have included performance metrics such as recall, precission, accuracy, etc., they have not included metrics such as MCC, Spe, Sen, etc., which we are used to seeing in such papers.

The authors include the sentence "This allowed calculating an average estimate of validation performance, using a 5-fold crossvalidation." but they do not include this in any of the metrics. In such a cv-5 situation, for example, the accuracy score would be expected to be given as mean +- std.

Reviewer #3: 1. Study Design and Sample Selection

Limited Scope of Anatomical Structures: The study exclusively focuses on the mandibular condyles, coronoid processes, and sigmoid notches for sex prediction. While these structures are relevant, the exclusion of other craniofacial bone structures that may also contribute to sex determination is a significant limitation. This narrow focus might reduce the generalizability and robustness of the predictive models.

Single Ethnic Group: The study’s sample is limited to Brazilian individuals, which introduces potential biases. The findings may not be applicable to other ethnic groups, limiting the external validity of the study. Without additional studies on diverse populations, the applicability of the results to a global context remains questionable.

2. Methodology and Data Collection

Imbalanced Age Representation: The sample includes individuals over ten years old, but the study does not address how age-related morphological changes might impact the models' predictive accuracy. Including a broad age range without specific stratification might introduce variability, potentially affecting the accuracy of sex prediction.

Potential Measurement Bias: The manual tracing of cephalometric measurements by assessors introduces a risk of measurement bias. Although the study reports high inter- and intra-examiner reliability, the subjective nature of these measurements could still impact the consistency and accuracy of the data.

3. Machine Learning Models and Statistical Analysis

Feature Selection Bias: The study employs a univariate analysis with a significance level of 10% for feature inclusion. This relatively high threshold increases the risk of including less relevant or noisy features, which may lead to overfitting or reduced model robustness. Additionally, relying solely on recursive feature elimination (RFECV) without exploring other feature selection methods could result in suboptimal feature sets.

Class Imbalance Handling: The use of the Synthetic Minority Over-sampling Technique (SMOTE) to address class imbalance is common, but it can sometimes generate synthetic samples that do not perfectly represent the minority class. This could lead to inflated model performance metrics, especially in scenarios with subtle differences between classes.

4. Interpretation of Results

Overemphasis on AUC: While the area under the curve (AUC) is a useful metric, the study places significant emphasis on it without equally considering other relevant metrics such as precision-recall curves, especially in the context of imbalanced datasets. AUC alone may not provide a complete picture of model performance, particularly in cases where the cost of false positives or false negatives is high.

Lack of External Validation: The models are evaluated using internal cross-validation and a hold-out test set, but there is no external validation with an independent dataset. This limits the ability to assess how well the models would perform in real-world scenarios or different populations, which is critical for forensic applications.

5. Discussion and Conclusion

Inadequate Discussion of Limitations: The discussion acknowledges some limitations, such as the focus on mandibular structures and the single-ethnicity sample, but it lacks depth in addressing how these limitations might specifically affect the study's findings. There is also insufficient discussion on potential ways to mitigate these limitations in future research.

Overstated Conclusions: The conclusion suggests that the mandibular characteristics and machine learning models can significantly contribute to sex prediction. However, given the study's limitations, this claim might be overstated. A more cautious interpretation, emphasizing the exploratory nature of the findings and the need for further validation, would be more appropriate.

6. Writing and Presentation

Clarity and Organization: The manuscript could benefit from clearer organization, particularly in the methods and results sections. Some sections contain dense information that could be better structured for readability and comprehension. Additionally, some technical terms are not adequately explained, which may limit the accessibility of the paper to non-specialist readers.

6. PLOS authors have the option to publish the peer review history of their article (what does this mean?). If published, this will include your full peer review and any attached files.

Reviewer #1: No

Reviewer #2: No

Reviewer #3: No

---

## [Author Response · Author response to Decision Letter 0]

10 Sep 2024

Ref: Submission ID PONE-D-24-17176

Dear Giulia Pascoletti,

Academic Editor,

PLOS ONE

Thank you very much for your message. We are submitting a revised version of our manuscript after addressing the areas of concern mentioned. The changes to the manuscript are highlighted in red within the document. Please find below our responses to the points raised in your email.

We hope that our corrections are appropriate and that the manuscript may now be reconsidered for publication. Should you have further questions or requests, please do not hesitate to contact us.

Yours sincerely,

Dr. Odilon Guariza Filho

Editor Comments:

Academic editor - In particular, reviewers have pointed out some lacks in the Introduction section that should be improved. Reviewers also recommended to increase the accuracy of the Methods section for what concern the metrics description. Discussion should better highlight limitations of the study, including also a more diffuse comparison with existing works.

Answer: We appreciate the comment, and all the points raised have been accepted and corrected in the text.

Reviewer Comments:

Reviewer #1 - 

1. The introduction section is too concise in my opinion. An overview of the methodologies adopted by other studies with the same objective should be added, as well as on the adopted ML techniques.

Answer: We appreciate the comment, and the suggestion has been accepted. The introduction has been expanded to include an overview of the methodologies adopted by other studies with the same objective. 

2. In the methods section I strongly recommend to be more accurate in explaining how the morphometric variables later employed were defined, also with the help of images

Answer: We appreciate the comment. Following this recommendation, the description has been enhanced to provide a more detailed explanation.

3. Did you check collinearity among the considered variables?

Answer: We appreciate the comment. In the case of the present study, collinearity was not a relevant concern, as only one numerical variable (Co-Gn) was included in the training of the models. We would also like to highlight that multicollinearity primarily affects linear models, such as logistic regression. Since the other models used, including the SVM with RBF kernel, do not rely on direct linear relationships between the variables, they are not significantly impacted by collinearity. Therefore, checking for collinearity was not necessary in the context of this work.

4. As the Introduction, the Discussion could contain a broader comparison with other similar studies.

Answer: We appreciate the comment. As suggested, a comparison with other similar studies has been added to the discussion.

Reviewer #2 - 

1. The introduction of the manuscript is quite inadequate. a more detailed literature review would have been expected from the authors.

Answer: We appreciate the comment, and the suggestion has been accepted. The introduction has been expanded to include an overview of the methodologies adopted by other studies with the same objective. 

2. "However, so far, no studies have been found in the literature that explore machine learning approaches using characteristics of the condyle, coronoid process, and sigmoid notch, whether morphometric or linear, for sex prediction." However, many features of mandible sex prediction have been successfully used and many ML models have been successfully introduced to the literature. This sentence requires serious clarification.

Answer: We appreciate the comment, and following this suggestion, the text has been revised to enhance its clarity.

3. Although the authors have included performance metrics such as recall, precission, accuracy, etc., they have not included metrics such as MCC, Spe, Sen, etc., which we are used to seeing in such papers.

Answer: Thank you for your comment. The choice of metrics was based on various studies in the field that use the same metrics, allowing for better comparison. All the studies cited that are similar to the present study employed the same metrics. This has been clarified in the methodology section for the reader.

4. The authors include the sentence "This allowed calculating an average estimate of validation performance, using a 5-fold crossvalidation." but they do not include this in any of the metrics. In such a cv-5 situation, for example, the accuracy score would be expected to be given as mean +- std.

Answer: Thank you for your suggestion. The metrics obtained during the cross-validation stage are available in Table 2. Following your suggestion, a new ROC curve specific to the cross-validation stage has been added to Figure 4. Additionally, all calculated metrics now include a 95% confidence interval, reflecting the variability and precision of the results obtained.

Reviewer #3 – 

1. Limited Scope of Anatomical Structures: The study exclusively focuses on the mandibular condyles, coronoid processes, and sigmoid notches for sex prediction. While these structures are relevant, the exclusion of other craniofacial bone structures that may also contribute to sex determination is a significant limitation. This narrow focus might reduce the generalizability and robustness of the predictive models.

Answer: We appreciate the comment. This point has been highlighted in the discussion section as a limitation of the study. Indeed, incorporating other craniofacial structures could further enhance the predictive power. However, focusing solely on these structures may serve as a valuable auxiliary method when other structures are not available. Additionally, no studies were found in the literature that focus exclusively on these specific structures.

2. Single Ethnic Group: The study’s sample is limited to Brazilian individuals, which introduces potential biases. The findings may not be applicable to other ethnic groups, limiting the external validity of the study. Without additional studies on diverse populations, the applicability of the results to a global context remains questionable.

Answer: We appreciate the comment. This point was highlighted in the discussion as a limitation of the study. However, it is important to emphasize that this is the first study to focus exclusively on these structures (condyle, sigmoid notch, and coronoid process) using supervised machine learning techniques.

3. Imbalanced Age Representation: The sample includes individuals over ten years old, but the study does not address how age-related morphological changes might impact the models' predictive accuracy. Including a broad age range without specific stratification might introduce variability, potentially affecting the accuracy of sex prediction.

Answer: This point was addressed in the discussion. The decision to include a broad age range was made because many criminal investigations involve victims of different ages, making it essential for identification and analysis methods to be applicable to a diverse age spectrum. Additionally, age estimation based on the available remains in some scenarios may often not be accurate. Although this decision may introduce variability in the results, particularly concerning the accuracy of the algorithms used, including a wide age range—from childhood to adulthood—enhances the external validity of the model when providing estimates in cases of identifying remains or analyzing bone evidence.

4. Potential Measurement Bias: The manual tracing of cephalometric measurements by assessors introduces a risk of measurement bias. Although the study reports high inter- and intra-examiner reliability, the subjective nature of these measurements could still impact the consistency and accuracy of the data.

Answer: We appreciate the comment, and this point has been acknowledged as a limitation of the present study.

5. Feature Selection Bias: The study employs a univariate analysis with a significance level of 10% for feature inclusion. This relatively high threshold increases the risk of including less relevant or noisy features, which may lead to overfitting or reduced model robustness. Additionally, relying solely on recursive feature elimination (RFECV) without exploring other feature selection methods could result in suboptimal feature sets.

Answer: We appreciate the comment. The choice of this significance level was intended to establish a conservative cutoff point, preventing the premature exclusion of variables with potential predictive importance. The recursive feature elimination with cross-validation (RFECV) technique was employed as a complement to the univariate analysis, specifically to minimize the risk of including a suboptimal set of variables.

6. Class Imbalance Handling: The use of the Synthetic Minority Over-sampling Technique (SMOTE) to address class imbalance is common, but it can sometimes generate synthetic samples that do not perfectly represent the minority class. This could lead to inflated model performance metrics, especially in scenarios with subtle differences between classes.

Answer: We appreciate the comment. In response to the suggestion, a paragraph on this point has been included in the discussion section. Although SMOTE may generate data that do not perfectly match the true distribution, in this study, the technique was applied to a large sample with no significant class imbalance. Moreover, the metrics obtained from 5-fold cross-validation were very similar to those from the test phase, suggesting that the model did not suffer from overfitting. This consistency indicates a reduced risk of distortion and a good capacity for generalization.

7. Overemphasis on AUC: While the area under the curve (AUC) is a useful metric, the study places significant emphasis on it without equally considering other relevant metrics such as precision-recall curves, especially in the context of imbalanced datasets. AUC alone may not provide a complete picture of model performance, particularly in cases where the cost of false positives or false negatives is high.

Answer: Thank you for your suggestion. In response, we have included the precision metric in the results and highlighted in the text the presentation of other metrics, along with their respective 95% confidence intervals, in Table 2.

8. Lack of External Validation: The models are evaluated using internal cross-validation and a hold-out test set, but there is no external validation with an independent dataset. This limits the ability to assess how well the models would perform in real-world scenarios or different populations, which is critical for forensic applications.

Answer: We appreciate the comment. This limitation has been added to the discussion section, with a recommendation that future studies evaluate the application of this model in real forensic scenarios to assess its external validity and performance across different populations.

9. Inadequate Discussion of Limitations: The discussion acknowledges some limitations, such as the focus on mandibular structures and the single-ethnicity sample, but it lacks depth in addressing how these limitations might specifically affect the study's findings. There is also insufficient discussion on potential ways to mitigate these limitations in future research.

Answer: We appreciate the comment. As previously noted, the limitations section has been restructured in accordance with the suggestions provided.

10. Overstated Conclusions: The conclusion suggests that the mandibular characteristics and machine learning models can significantly contribute to sex prediction. However, given the study's limitations, this claim might be overstated. A more cautious interpretation, emphasizing the exploratory nature of the findings and the need for further validation, would be more appropriate.

Answer: Thank you for your comment. As suggested, the conclusion has been revised to highlight the study's limitations and to adopt a more cautious interpretation of the findings.

11. Clarity and Organization: The manuscript could benefit from clearer organization, particularly in the methods and results sections. Some sections contain dense information that could be better structured for readability and comprehension. Additionally, some technical terms are not adequately explained, which may limit the accessibility of the paper to non-specialist readers.

Answer: Thank you for your comment. Following the suggestion, the main aspects of the methodology, particularly those related to the variables and metrics, have been clarified to enhance readability for the reader.

---

## [Decision Letter · Decision Letter 1]

15 Oct 2024

SEX PREDICTION THROUGH MACHINE LEARNING UTILIZING MANDIBULAR CONDYLES, CORONOID PROCESSES, AND SIGMOID NOTCHES FEATURES

PONE-D-24-17176R1

Dear Dr. Guariza-Filho,

We’re pleased to inform you that your manuscript has been judged scientifically suitable for publication and will be formally accepted for publication once it meets all outstanding technical requirements.

The comments and concerns raised by the reviewers have been addressed, and the manuscript was improved following the provided indications.

Kind regards,

Giulia Pascoletti, Ph.D.

Academic Editor

PLOS ONE

Additional Editor Comments (optional):

Reviewers' comments:

Reviewer's Responses to Questions

**Comments to the Author**

1. If the authors have adequately addressed your comments raised in a previous round of review and you feel that this manuscript is now acceptable for publication, you may indicate that here to bypass the “Comments to the Author” section, enter your conflict of interest statement in the “Confidential to Editor” section, and submit your "Accept" recommendation.

Reviewer #1: All comments have been addressed

Reviewer #2: (No Response)

Reviewer #3: (No Response)

2. Is the manuscript technically sound, and do the data support the conclusions?

Reviewer #1: Yes

Reviewer #2: Partly

Reviewer #3: (No Response)

3. Has the statistical analysis been performed appropriately and rigorously? 

Reviewer #1: Yes

Reviewer #2: No

Reviewer #3: (No Response)

4. Have the authors made all data underlying the findings in their manuscript fully available?

Reviewer #1: No

Reviewer #2: No

Reviewer #3: (No Response)

5. Is the manuscript presented in an intelligible fashion and written in standard English?

Reviewer #1: Yes

Reviewer #2: Yes

Reviewer #3: (No Response)

6. Review Comments to the Author

Reviewer #1: (No Response)

Reviewer #2: The authors have organized some requests. It is a study lacking hyperparametrization and should be re-evaluated with a better methodology.

Reviewer #3: Accept

Please use the space provided to explain your answers to the questions above. You may also include additional comments for the author, including concerns about dual publication, research ethics, or publication ethics.

7. PLOS authors have the option to publish the peer review history of their article (what does this mean?). If published, this will include your full peer review and any attached files.

Reviewer #1: **Yes: **Alessandra Aldieri

Reviewer #2: No

Reviewer #3: No

---

## [Editor Report · Acceptance letter]

6 Nov 2024

PONE-D-24-17176R1 

PLOS ONE

Dear Dr. Guariza-Filho, 

I'm pleased to inform you that your manuscript has been deemed suitable for publication in PLOS ONE. Congratulations! Your manuscript is now being handed over to our production team.

Kind regards, 

on behalf of

Dr. Giulia Pascoletti 

Academic Editor

PLOS ONE